# Design of a Compliant Mechanism Based Four-Stage Amplification Piezoelectric-Driven Asymmetric Microgripper

**DOI:** 10.3390/mi11010025

**Published:** 2019-12-24

**Authors:** Xiaodong Chen, Zilong Deng, Siya Hu, Jinhai Gao, Xingjun Gao

**Affiliations:** 1College of Mechanical Engineering, Liaoning University of Petroleum and Chemical Technology, Fushun 113001, China; 2College of Mechanical Engineering and Automation, Northeast University, Shenyang 110000, China

**Keywords:** microgripper, design, finite element analysis (FEA), piezoelectric drive, experimental verification

## Abstract

The existing symmetrical microgrippers have larger output displacements compared with the asymmetrical counterparts. However, the two jaws of a symmetrical microgripper are less unlikely to offer the same forces on the two sides of a grasped micro-object due to the manufacture and assembly errors. Therefore, this paper proposes a new asymmetric microgripper to obtain stable output force of the gripper. Compared with symmetrical microgrippers, asymmetrical microgrippers usually have smaller output displacements. In order to increase the output displacement, a compliant mechanism with four stage amplification is employed to design the asymmetric microgripper. Consequently, the proposed asymmetrical microgripper possesses the advantages of both the stable output force of the gripper and large displacement amplification. To begin with, the mechanical model of the microgripper is established in this paper. The relationship between the driving force and the output displacement of the microgripper is then derived, followed by the static characteristics’ analysis of the microgripper. Furthermore, finite element analysis (FEA) of the microgripper is also performed, and the mechanical structure of the microgripper is optimized based on the FEA simulations. Lastly, experimental tests are carried out, with a 5.28% difference from the FEA results and an 8.8% difference from the theoretical results. The results from theoretical calculation, FEA simulations, and experimental tests verify that the displacement amplification ratio and the maximum gripping displacement of the microgripper are up to 31.6 and 632 μm, respectively.

## 1. Introduction

With the rapid development of micro-/nano-technologies, precision processing, biological engineering, microelectronics, and aerospace [1,2,3,4,5], micro-manipulators with better operating performance are desired. As micro-operated end effectors, microgrippers are placed at the end of the arms of manipulators to interact with micro-objects, so they play a vital role in determining the success of micro-operation tasks. Compared with traditional rigid mechanisms, compliant mechanisms are more suitable for the design of microgrippers, due to their advantages such as no backlash, no requirement for lubrication, simplified manufacture, and low part count [6,7]. At present, a variety of microgrippers has been developed with different driving modes. The common driving modes of microgrippers are the piezoelectric drive [8,9], electrostatic drive [10,11], electrothermal drive [12,13], shape memory alloy (SMA) drive [14,15], pneumatic drive [16,17], etc. Compared with other driving modes, the piezoelectric drive has the advantages of small size, large output force, high sensitivity, and no gap. Therefore, it is more widely used as the actuators of microgrippers [18,19].

It is found from the current studies that the aim of the development of microgrippers is to increase the displacement amplification and improve the motion accuracy. The output displacement of a piezoelectric actuator is too small, usually tens of microns. In order to enlarge the strokes of piezoelectric actuators, displacement amplifiers are widely employed in piezoelectric driven systems. Existing micro-displacement amplifiers include the lever amplifier, bridge amplifier, and rhombus amplifier. Displacement amplifiers consist of a single lever, bridge, or rhombic and are categorized as single stage amplifiers. Amplifiers composed of multiple single stage amplifiers are called multi-stage amplifiers [20]. A multi-stage amplifier can largely increase the clamping stroke of a microgripper. The gripping accuracy of a microgripper mainly involves two issues: kinematic accuracy and output force of the gripper accuracy. The kinematic accuracy is mainly manifested by parallel gripping accuracy. When the gripping object is an irregularly, such as spherical or cylindrical, shaped component, the parallel gripping can realize stable gripping and assembly [21,22]. Parallel gripping can also reduce the stress concentration on the gripper when the gripping objects are fragile chip devices [23]. The accuracy of the output force of the gripper mainly depends on the control of the output force of the gripper of the grasping jaws. The existing gripping methods of microgrippers are symmetrical gripping and asymmetric gripping. Symmetrical gripping easily to damages the micro-object due to the uneven forces exerted on the micro-object from the left and right jaws of the gripper. One grasping jaw of an asymmetric microgripper is fixed to the ground, and the other grasping jaw is moveable. Therefore, the piezoelectric actuation system of a piezoelectric driven asymmetric microgripper is used to control the motion of one of the grasping jaws, which enables the motion control to be more accurate and stable. However, the output displacement of an asymmetric microgripper is small, which is usually half of the clamping stroke of the symmetrical counterpart.

Li et al. [24] designed a single stage symmetrical microgripper based on the principle of lever amplification to achieve a desired displacement amplification, but could not achieve parallel gripping. Cui et al. [25] designed a symmetrical microgripper based on the principle of lever amplification that could realize parallel gripping. However, the actual maximum output forces of the left and right jaws were different, 8.02 mN and 9.24 mN, respectively. The output forces were different, so it was easy to destroy the microparts in the clamping process. Sun et al. [26] designed a two stage symmetrical microgripper based on the principle of lever amplification and triangle amplification, which realized parallel clamping of the jaws and high amplification. Wang et al. [27] designed a three stage symmetrical microgripper based on the principle of lever amplification and triangle amplification to further improve the amplification. However, the microgrippers designed by Sun et al. [26] and Wang et al. [27] could not achieve stable gripping. Koo et al. [28] designed a single stage asymmetric microgripper based on the principle of lever amplification, which realized the stable gripping of the jaws, but could not be clamped in parallel. Xing et al. [29] designed a single stage asymmetric microgripper based on the principle of lever amplification, which realized the parallel gripping of the jaws, but the amplification was small. Zhao et al. [30] designed a two stage asymmetric microgripper based on the principle of lever amplification, which realized the stable gripping of the jaws and improved displacement amplification. Based on the principle of lever amplification and triangle amplification, Liang et al. [31] designed a three stage asymmetric microgripper, which realized the parallel gripping, but the amplification was smaller than that of Li et al. [24], Sun [26], and Wang [27].

In summary, compared with symmetrical microgrippers, an asymmetric microgripper has more stable gripping performance. It is necessary to design a kind of piezoelectric driven microgripper with high displacement amplification and high stable gripping performance. In this paper, a new piezoelectric driven asymmetric microgripper is proposed with a high amplification ratio (31.6) and parallel gripping performance.

## 2. Structure Design and Motion Principle of the Microgripper

### 2.1. Structure Design of the Microgripper

Figure 1 shows a diagram of the main view of the microgripper. The size of the mechanism was 68.8 mm × 34.55 mm × 5 mm. The microgripper mainly consisted of a stacked piezoelectric (PZT) ceramic actuator (SPCA), an asymmetric right-circular flexure hinge, a rectangular flexure hinge, a grasping jaw, a double lever amplifier, a double bridge amplifier, a fixing hole, and a preload bolt. In order to further improve the clamping stroke and obtain larger output displacement, a four stage amplifier was designed, including a double bridge amplifier and a double lever amplifier. One end of SPCA was fixed with the displacement transmission mechanism (DTM) by the preload bolt, and the preload force on SPCA could be adjusted by the bolt. The double bridge amplifier and the lever amplifier were directly connected in series to obtain a larger displacement amplification. Rectangular hinges were employed in the double bridge amplifier due to the fact that rectangular hinges have large displacements, but small stress concentrations compared with other types of hinges [32,33,34].

### 2.2. Movement Principle of the Microgripper

Figure 2 shows the structure diagram of the microgripper, and the corresponding motion principle is shown in the figure. The microgripper consisted of two bridge amplifiers and two lever amplifiers.

Firstly, the bridge amplification mechanism was analyzed. The bridge type amplifier was a symmetrical structure, and the deformation of four side lengths was the same. The stress analysis of one side length was conducted, and the corresponding force analysis diagram is shown in Figure 3. The upper left side length *AB* of the double bridge amplifier in Figure 2 was analyzed. After the piezoelectric ceramic generated horizontal thrust, point *A* moved to the left along the horizontal direction to point *A*′, and point *B* moved to the left along the vertical direction at *B*′; if the moment of side length *AB* was counterclockwise, then the moment of side length *AB* was 2*M*_r_.

According to the stress relationship, the formula can be achieved:(1)$$\left\{ \begin{array}{l} F_{Ax}=F_{Bx}=F_{x}=F_{PST}/4\\ F_{Ay}=F_{By}=F_{y}=F_{PST}/4\\ 2M_{r}=F_{x}L_{AB}\sin\alpha +F_{y}L_{AB}\cos\alpha \end{array}\right.$$

A point on side length *AB* was selected. The distance from this point to point *A* was *c*, and the corresponding force and moment equations could be obtained: (2)$$\left\{ \begin{array}{l} F_{N}=F_{x}\cos\alpha -F_{y}\sin\alpha \\ M=M_{r}-F_{x}c\sin\alpha -F_{y}c\cos\alpha \end{array}\right.$$

The deformation energy of side length *AB* could be divided into two parts: tensile deformation energy *E*_1_ and bending deformation energy *E*_2_; thus:(3)$$E=E_{1}+E_{2}$$

According to the second theorem of Castigliano, the deformation of the side length in the x direction can be written as:(4)$$\Delta x=\Delta x_{1}+\Delta x_{2}$$
(5)$$\begin{array}{l} \Delta x=\frac{\partial E}{\partial F_{x}}=\frac{F_{N}L_{AB}}{EA}\frac{\partial F_{N}}{\partial F_{x}}+\int_{0}^{L_{AB}}\frac{M}{EI}\frac{\partial M}{\partial F_{x}}dc=\frac{L_{AB}}{EA}(F_{x}\cos\alpha -F_{y}\sin\alpha )\cos\alpha +\\ \frac{L_{AB}{}^{3}}{12EI}\left(F_{x}\sin\alpha +F_{y}\cos\alpha \right)\sin\alpha \end{array}$$
where *EA* is tensile stiffness and *EI* is bending stiffness.
(6)$$\begin{array}{l} \Delta y=\frac{\partial E}{\partial F_{y}}=\frac{F_{N}L_{AB}}{EA}\frac{\partial F_{N}}{\partial F_{y}}+\int_{0}^{L_{AB}}\frac{M}{EI}\frac{\partial M}{\partial F_{y}}dc=\frac{L_{AB}}{EI}\left(F_{x}\cos\alpha -F_{y}\sin\alpha \right)\sin\alpha +\\ \frac{L_{AB}^{3}}{12EI}\left(F_{x}\sin\alpha +F_{y}\cos\alpha \right)\cos\alpha \end{array}$$

The displacement amplification ratio of single bridge amplifier can be written as:(7)$$R_{\mathrm{amp}0}=\frac{\Delta y}{\Delta x}=\frac{\left(L_{AB}{}^{2}-h^{2}\right)\sin\alpha \cos\alpha }{h^{2}\cos^{2}\alpha +6L_{AB}{}^{2}\sin^{2}\alpha }$$

Thus, the displacement amplification ratio of double bridge amplifier is:(8)$$R_{\mathrm{amp}1}=\frac{2\Delta y}{\Delta x}=\frac{2\left(L_{AB}{}^{2}-h^{2}\right)\sin\alpha \cos\alpha }{h^{2}\cos^{2}\alpha +6L_{AB}{}^{2}\sin^{2}\alpha }$$

The output stage of the compliant parallelogram parallel mechanism on the moveable jaw had a parasitic rotation about the Z-axis, which reduced the gripping accuracy of the jaw [35,36,37]. However, the parasitic rotation was much smaller than the principal motion, the translation along the X-axis. It could be calculated from finite element analysis that the rotation displacement was about 0.0658% of the gripping displacement, as shown in Figure 4. Therefore, the parasitic rotation could be ignored in this case, while it could also be eliminated by applying the actuation force along the stiffness center of the compliant parallelogram mechanism (shown in Figure 4b) in other cases if a larger geometric dimension was not a problem.

Based on the pseudo rigid body model (PRBM) method, the analysis of the double lever amplifier was carried out. The PRBM of the mechanism is shown in Figure 5. Displacement *d*_in_ was applied to the input end, and the first displacement amplification was realized by the lower lever amplifier. The output displacement *s* of the lower lever amplifier can be expressed as:(9)$$d_{s}=\frac{\left(l_{2}+l_{3}\right)d_{in}}{l_{1}}$$

The movement of the lower lever amplifier drove the upper lever amplifier to undergo secondary deformation. When the mechanism was not deformed, the angle between the line *OA* and the horizontal direction was θ. After the mechanism was deformed, the change in the amount of the angle θ was Δθ. Then, there was the following geometric relationship:(10)$$d_{in}=\frac{l_{1}\sqrt{l_{3}^{2}+l_{4}^{2}}}{l_{2}+l_{3}}\left[\cos\left(\theta \right)-\cos\left(\theta +\Delta \theta \right)\right]$$
(11)$$d_{out}=l_{5}\left[\sin\left(\theta +\Delta \theta \right)-\sin\left(\theta \right)\right]$$

In the process of movement, the flexure hinge produced small deformation, and the maximum deformation was usually no more than one thousandth of its own size; then Δθ≈0, so the displacement amplification of the mechanism can be expressed as:(12)$$R_{\mathrm{amp}2}=\frac{d_{out}}{d_{in}}=\frac{l_{5}\left(l_{2}+l_{3}\right)}{l_{1}\sqrt{l_{3}^{2}+l_{4}^{2}}}$$

Therefore, the amplification of the four stage amplification microgripper is:
(13)$$R_{\mathrm{amp}}=R_{\mathrm{amp}1}R_{\mathrm{amp}2}=\frac{2\Delta yd_{out}}{\Delta xd_{in}}=\frac{2l_{5}\left(l_{2}+l_{3}\right)\left(L_{AB}{}^{2}-h^{2}\right)\sin\alpha \cos\alpha }{l_{1}\sqrt{l_{3}^{2}+l_{4}^{2}}\left(h^{2}\cos^{2}\alpha +6L_{AB}{}^{2}\sin^{2}\alpha \right)}$$

## 3. Finite Element Analysis

### 3.1. Size Parameter Optimization

The structure diagram of the microgripper is shown in Figure 1. The performance of the microgripper was mainly determined by the structure size of the hinges. Based on the ANSYS software response surface method, the structure size of the microgripper hinges was optimized. In order to obtain larger displacement amplification, the output displacement was chosen as the optimization objective function. The key parameters *a*, *b*, *c*, *d*, *e*, *f*, *g*, *h* were selected as input variables, which were defined according to practical application and manufacturing conditions within a certain range. The maximum stress in the working process should be less than the yield strength of the material. In order to improve the performance of the microgripper, the optimization design was carried out as follows:

(1) Objective: Maximum output displacement.

(2) Related parameters: *n*, *t*_3_, *b*_l_, *b_w_*, *m*, *t_2_*, *d*, *e*, *f*, *g*, *r*, *t*_1_.

(3) Subject to:

(a) constraint equations: $\sigma_{\max}=\frac{\left(\sigma \right)}{n_{a}}$, where (σ) denotes the tensile yield stress and $n_{a}=2$ represents the factor of safety;

(b) parameter range: 0.4 mm ≤ *n* ≤ 0.8 mm, 0.2 mm ≤ *t*_3_ ≤ 0.4 mm, 6 mm ≤ *b_l_* ≤ 8 mm, 1.5 mm ≤ *b_w_* ≤ 2.5 mm, 2.5 mm ≤ *m* ≤ 3.5 mm, 0.2 mm ≤ *t*_2_ ≤ 0.4 mm, 14 mm ≤ *d* ≤ 18 mm, 20 mm ≤ *e* ≤ 25 mm, 3 mm ≤ *f* ≤ 5 mm, 10 mm ≤ *g* ≤ 12 mm, 0.8 mm ≤ *r* ≤ 1.8 mm, 0.2 mm ≤ *t*_1_ ≤ 0.4 mm.

The optimal geometric parameters of the microgripper could be obtained by looking at the trade-off diagram and achieve the optimal performance. The optimized size is shown in Table 1.

### 3.2. Performance Analysis of Microgripper

The finite element analysis software ANSYS (15.0) was used to simulate the deformation of the microgripper. The material of the microgripper and micro-parts in the analysis process was 7075 aluminum alloy, a modulus of elasticity E = 71 GPa, a Poisson’s ratio ν = 0.33, a yield strength σ = 455 MPa, and a density ρ = 2810 kg/m^3^. In the process of simulation, the 3D solid model built in SolidWorks was imported into ANSYS for mesh generation; constraints and static loads were applied; and static analysis was carried out. For the flexible part of the flexure hinges of the microgripper, we set a small mesh generation parameter (0.1 mm), and the other parts generated the mesh automatically, which could not only accurately describe these parameters to improve the accuracy of the analysis results, but also accelerate the analysis speed and save time; the constraint conditions and static load application position are shown in Figure 6a.

Figure 6b,c is the corresponding displacement nephogram of the microgripper when 20 μm of input displacement was applied to the input end and the microparts did not grip. Figure 6b is the displacement nephogram of the microgripper. It can be clearly seen that during the closing process of the microgripper, the right jaw gripped in parallel; the maximum output displacement of one side was 656.43 μm; the displacement amplification was 32.82 times. Figure 6c is the strain nephogram of the microgripper, from which it can be seen that the maximum stress was 344.5 MPa, which was less than the yield strength of the material, so the product could be used safely.

The grasping force was estimated by FEA. Figure 7 shows the SPCA input force versus the output force of the gripper when the micro-axes of 600 μm, 450 μm, and 300 μm were gripped by the microgripper, respectively, where *D* is the initial distance between the jaws, *d* is the micro-object diameter, and *s* is the distance between the jaw and micro-objects. It can be seen from the figure that the linearity of the input displacement and the output force was large and easy to control, and a smooth transition was achieved from the closing of the grasping jaw to the gripping of the micro-axis. When the micro-axis of 600 μm was gripped, the theoretical output force of the gripper jaw was 270 mN under the input displacement of 22 N.

## 4. Experiments

### 4.1. Physical Model of Microgripper

The physical model of the microgripper is shown in Figure 8. The material used for the microgripper was 7075-T6 (SN) aluminum alloy. The microgripper was processed by the wire electrical-discharge machining (WEDM) with slow walking. After processing, the microgripper was drilled and polished. The experimental equipment included the HPV-1 C 0300 A0300 piezoelectric ceramic driving power supply (Suzhou Mat Inc., Suzhou, China), micro-sodium positioning worktable (Suzhou Mat Inc., Suzhou, China), PZT (SZBS150/5 × 5/20, open-loop travel 20 μm, Suzhou Mat Inc., Suzhou, China) driving micro-positioning stage (Suzhou Mat Inc., Suzhou, China), high resolution capacitive displacement sensor (BJZD Company MA-0.5, Beijing, China), data acquisition card (NI Company PCI-6221, Dongwan, China), 24 V DC regulated power supply WP100-D-G (Suzhou Mat Inc., Suzhou, China), host computer and display matched with the sensor signal amplifier BSQ-2 (Suzhou Mat Inc., Suzhou, China), and 24 V DC regulated power supply WP100-D-G. In order to eliminate the external interference as much as possible, all the devices were installed on the high performance vibration isolation platform (ZPT-G-M, zh-gezhen, Wuhan, China).

### 4.2. Experimental Verification

In order to further test the performance of the microgripper, a series of experiments was carried out. The experimental device is shown in Figure 9. Figure 10a is the functional block diagram of the experimental device, and Figure 10b is the actual working diagram of the experimental device.

Figure 11 shows the relationship between the input displacement and the piezoelectric actuator driving voltage from 0 to 150 V. The results confirmed the nonlinear characteristic and hysteresis of the piezoelectric material. When the input voltage was 150 V, the maximum output displacement was 22.08 μm, which was larger than the nominal output displacement of SPCA. Under the open-loop control condition, when the driving voltage was 20 V, the output displacement difference of expansion and contraction reached the maximum value, which was 2.74 μm.

However, the nonlinearity of the piezoelectric material did not affect the linear relationship between the input and output of the microgripper. Figure 12 shows the SPCA input displacement versus the tip displacement of the grasping jaw. From the graph, it can be seen that the input displacement of SPCA was linearly related to the tip displacement of the grasping jaw, which showed that the microgripper had a stable performance.

The theoretical amplification factor was 34.65 times; the FEA simulation amplification factor was 32.82 times; and the experimental amplification factor was 31.6 times. The error between the experiment and the FEA simulation was 5.28%, while the error between the experiment and the theoretical calculation was 8.80%. The three displacement amplification ratios by different methods were very close, which proved the correctness of the theoretical calculation. There was a certain degree of deviation between the physical model and the theoretical modeling due to the machining error, which resulted in the error between the experiment value, FEA simulation value, and theoretical calculation value. The idea of theoretical calculation is based on geometric relation, which only considers the deformation of the flexure hinges and regards the other parts of the flexure hinge as rigid bodies. Actually, there was indeed deformation in other parts of the flexure hinges. On the other hand, the multi-stage amplifier could restrain the single machine amplifier; that is why the theoretical calculation value was larger than the FEA simulation value and the experiment value. The accuracy error in the process of microgripper machining, the measurement error in the process of the experiment, and the influence of equipment vibration and noise were inevitable; that was why the experiment value was smaller than the FEA simulation value.

The experimental results of displacement amplification are shown in Table 2. The input displacement and output displacement were measured by a displacement sensor. In order to further verify the performance of the microgripper, the test device was started and closed five times. Under the action of 150 V driving voltage, the input displacement of SPCA and the tip displacement of grasping jaw were measured five times. The average value of the five displacement magnifications *k* = 31.6, and the standard deviation was 0.019. The experimental results proved the stability of the microgripper.

In order to verify the applicability of the microgripper in the operation of micro-objects with different shapes and sizes, several grasping experiments were carried out. Figure 13 shows the microgripper grasping metal plate, wire, and plastic plate, respectively. The results showed that the microgripper had the characteristics of parallel gripping and a large grasping range for different shapes of objects and could successfully grasp micro-objects. Figure 14 shows the microgripper parallel grasping a 250 μm metal ball.

### 4.3. Performance Comparison

The amplification of our microgripper versus similar devices [13,14,15,16,17,18,19] is shown in Table 3. The amplification of the microgripper designed in this paper reached 31.6 times, which was greater than other microgrippers.

## 5. Conclusions

To address the disadvantages of the existing symmetrical microgrippers, this paper proposed a new PZT actuated microgripper. A large displacement amplification ratio was achieved for the proposed microgripper by using a four stage flexure based amplification. The new microgripper not only had large output displacement, the same as the traditional symmetrical microgrippers, but also possessed stable gripping performance. The relationship between the input and output displacements of the microgripper was obtained through theoretical derivation and verified by FEA simulations. A prototype of the microgripper was manufactured by WEDM with slow walking, and experimental tests were carried out on the prototype. The experimental results matched well with both the theoretical results and the simulation results. The microgripper designed in this paper could achieve high magnification displacement up to 31.6 and could operate on micro-parts of any irregular shapes smaller than 632 μm in size. In this paper, a series of gripping experiments was performed on the microgripper, and a variety of irregularly shaped micro-parts were successfully gripped without any damage. The design principle of the compliant mechanism based microgripper provided a useful reference for research on multi-stage amplification manipulators and microgrippers.

## Figures and Tables

**Figure 1 micromachines-11-00025-f001:**
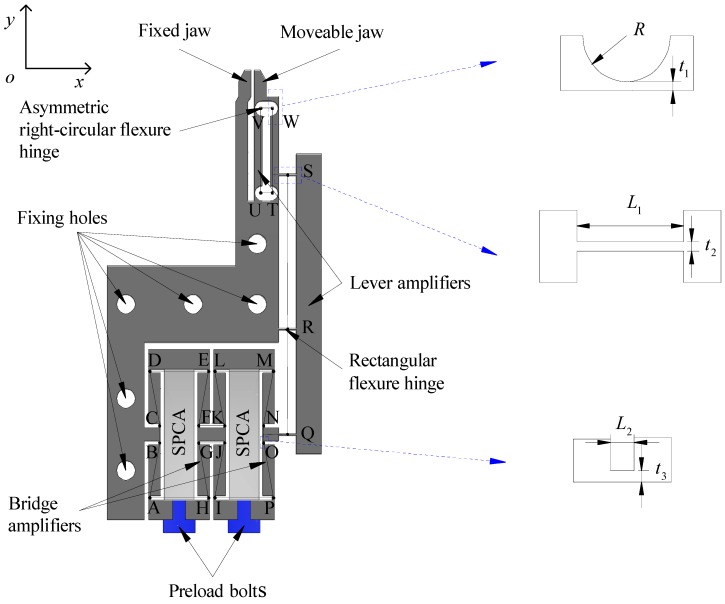
Main view of the microgripper. SPCA, stacked piezoelectric (PZT) ceramic actuator.

**Figure 2 micromachines-11-00025-f002:**
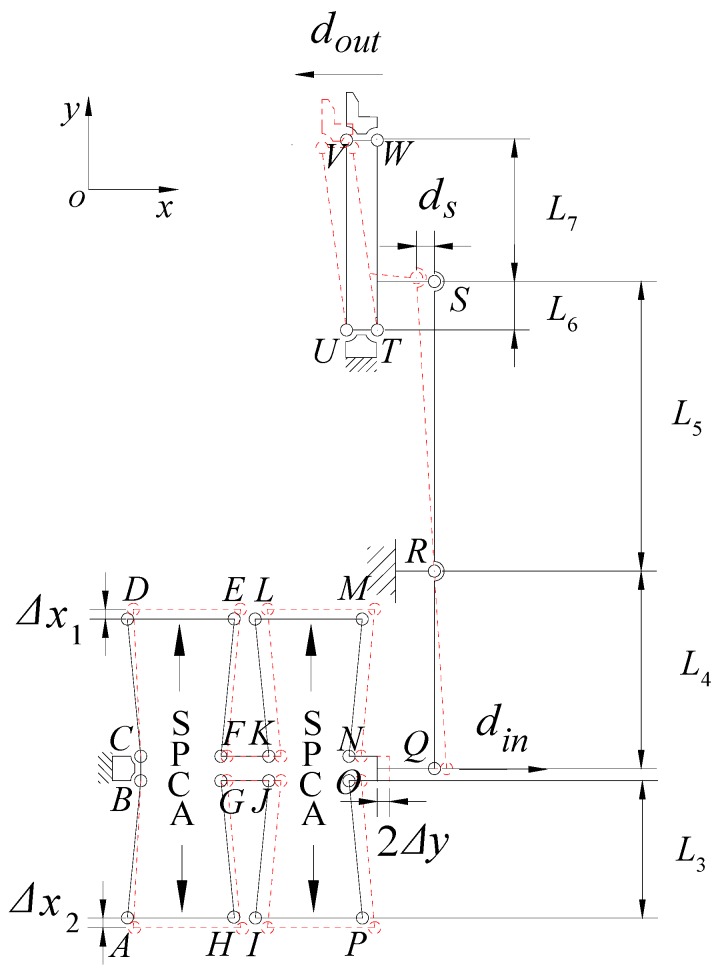
Pseudo rigid body model of the PZT driven microgripper.

**Figure 3 micromachines-11-00025-f003:**
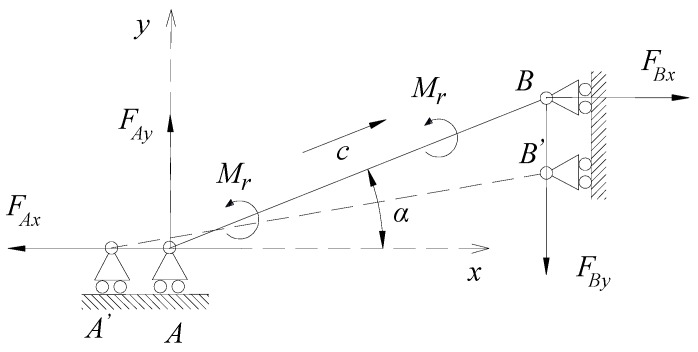
Single side deformation of the bridge amplifier.

**Figure 4 micromachines-11-00025-f004:**
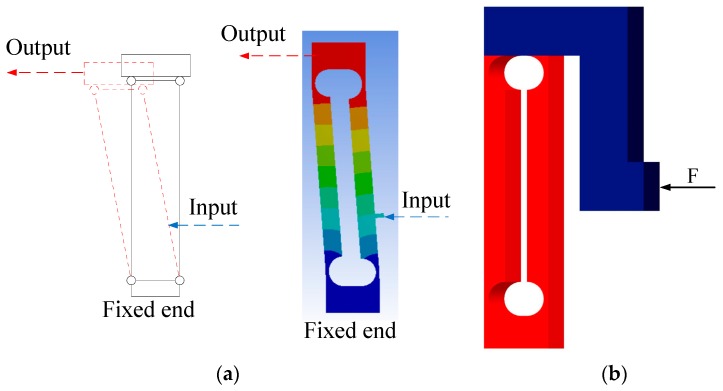
Amplification principle of the parallelogram mechanism: (**a**) traditional parallelogram mechanism and (**b**) a new type of parallelogram mechanism.

**Figure 5 micromachines-11-00025-f005:**
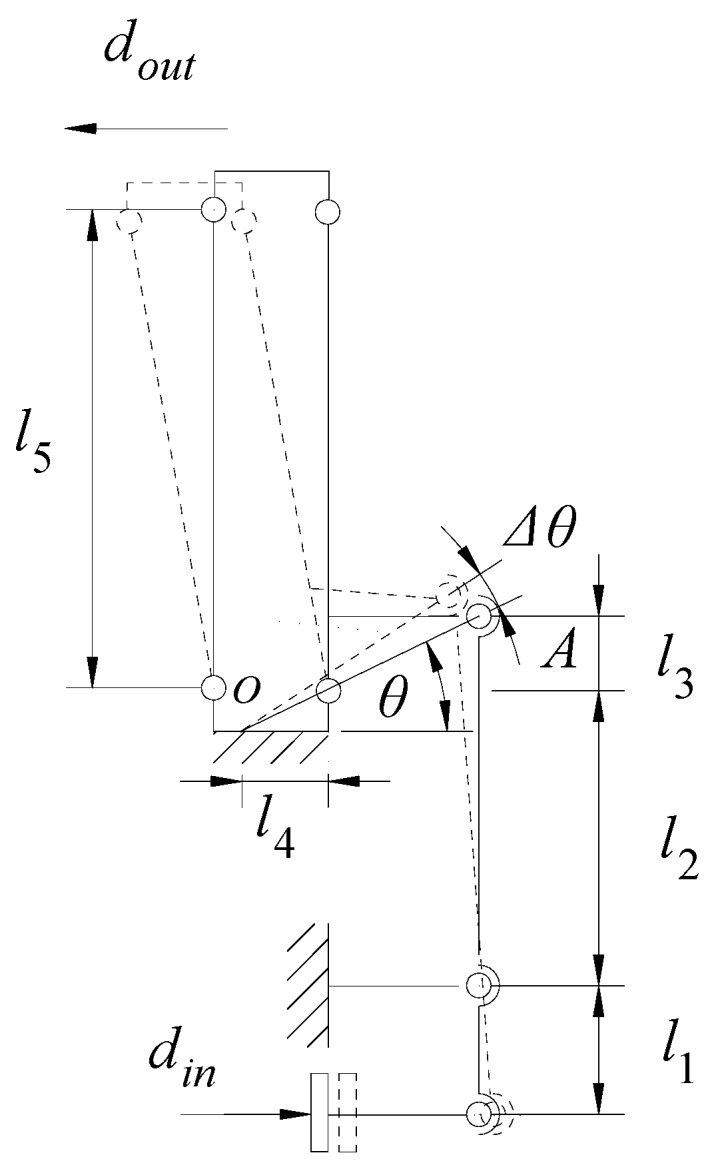
Pseudo rigid body model (PRBM) of double lever amplifiers.

**Figure 6 micromachines-11-00025-f006:**
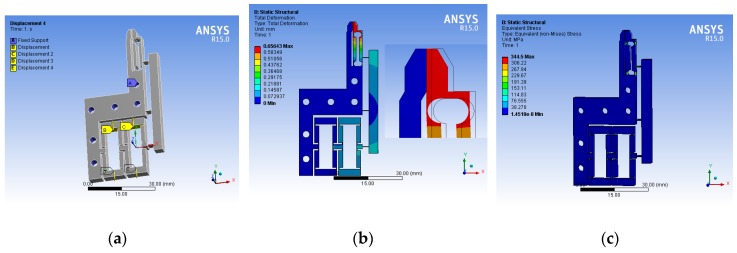
FEA diagram of the microgripper: (**a**) FEA settings, (**b**) deformation pattern and (**c**) stress pattern.

**Figure 7 micromachines-11-00025-f007:**
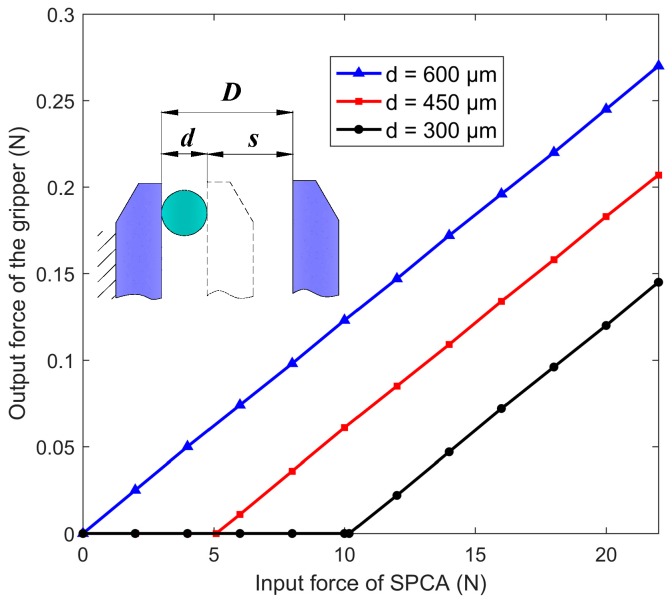
SPCA input force versus the output force of the gripper.

**Figure 8 micromachines-11-00025-f008:**
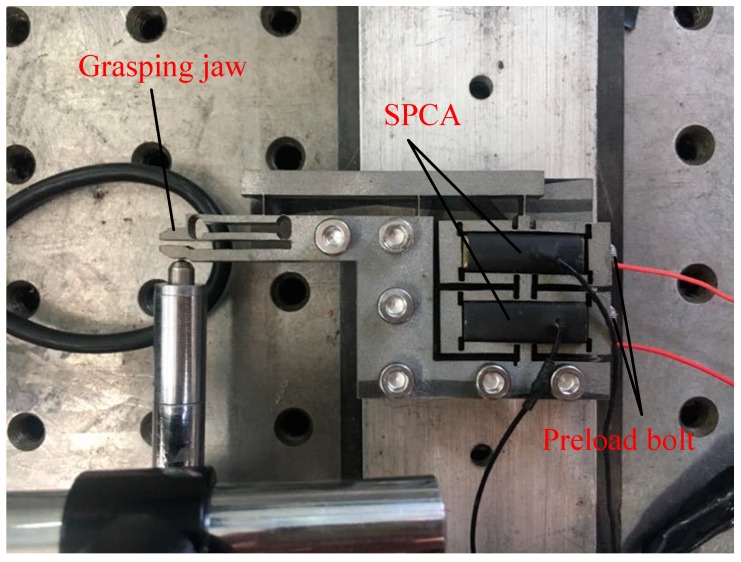
Physical model of the microgripper.

**Figure 9 micromachines-11-00025-f009:**
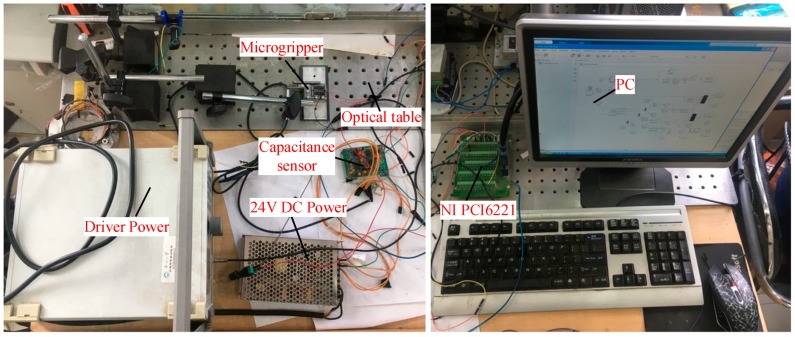
Experimental setup of the microgripper.

**Figure 10 micromachines-11-00025-f010:**
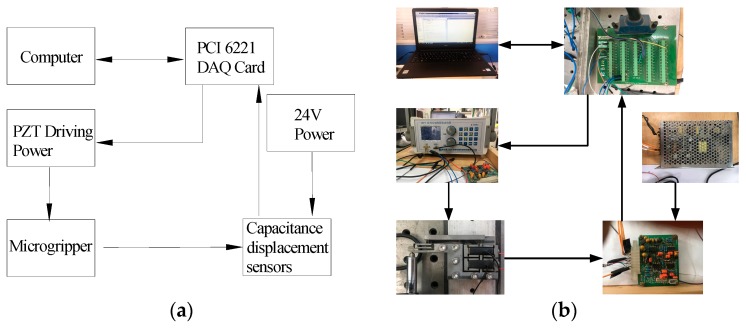
Working diagram of the experimental device. (**a**) functional block diagram and (**b**) actual working diagram.

**Figure 11 micromachines-11-00025-f011:**
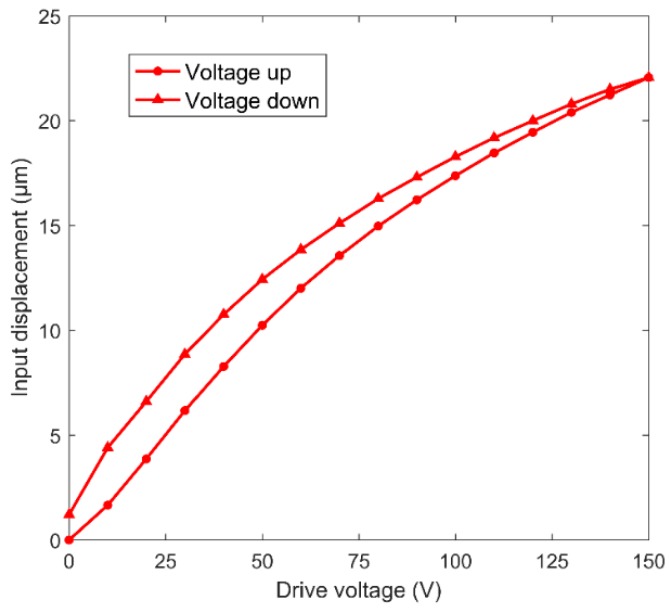
Input displacement versus the drive voltage.

**Figure 12 micromachines-11-00025-f012:**
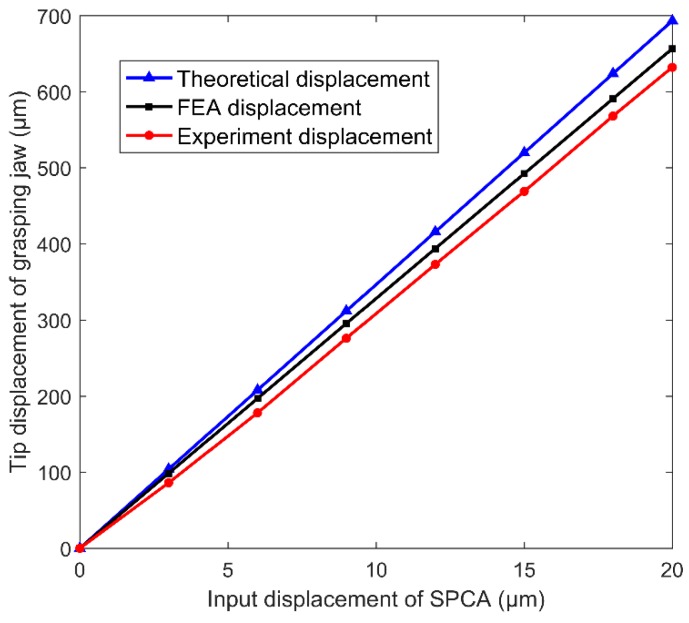
SPCA input displacement versus the tip displacement of the grasping jaw.

**Figure 13 micromachines-11-00025-f013:**
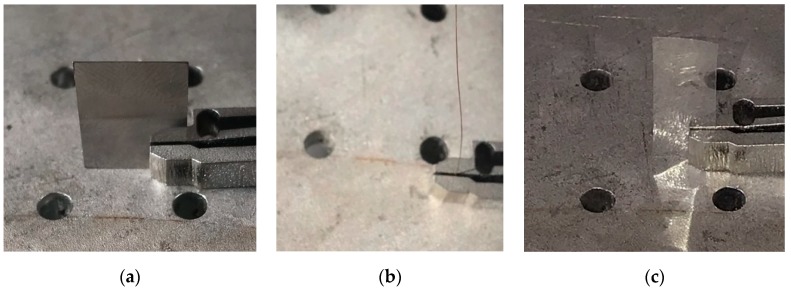
Grasping manipulation for different shaped and sizes of micro-objects (from left to right): (**a**) 300 μm metal plate, (**b**) 250 μm metal wire and (**c**) 120 μm plastic plate.

**Figure 14 micromachines-11-00025-f014:**
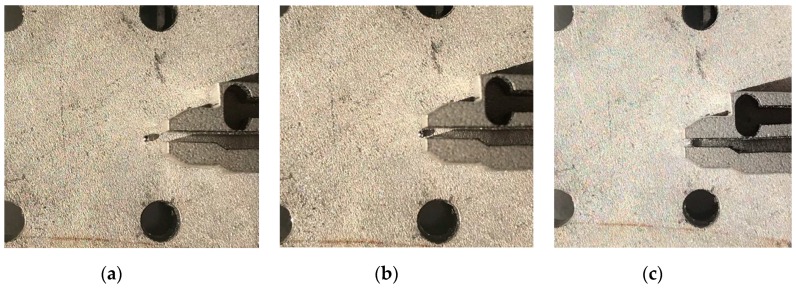
(**a**,**b**) Approaching and (**c**) grasping the micro-object.

**Table 1 micromachines-11-00025-t001:** Optimized dimensions.

Parameter	Description	Value
*L*_2_ × *t*_3_	Dimensions of rectangular flexure hinges	0.64 × 0.28 mm^2^
*L* _3_	Dimensions of the side length of the bridge amplifier	7.72 × 1.88 mm^2^
*L*_1_ × *t*_2_	Dimensions of rectangular flexure hinges	2.87 × 0.28 mm^2^
*L* _4_	Dimension of the lever short arm	16.12 mm
*L* _5_	Dimension of the lever long arm	23.64 mm
*L* _6_	Dimension of the lever short arm	3.96 mm
*L* _7_	Dimension of the lever long arm	11.37 mm
*R* × *t*_1_	Dimension of asymmetric right-circular flexure hinges	1.18 × 0.28 mm^2^

**Table 2 micromachines-11-00025-t002:** Experiment results of displacement amplification.

No.	Driven Power (V)	Input Displacement (μm)	Tip Displacement (μm)	*k*
1	150	22.06	697.1	31.60
2	150	21.93	693.65	31.63
3	150	22.15	699.94	31.60
4	150	22.18	700.44	31.58
5	150	21.97	694.03	31.59

**Table 3 micromachines-11-00025-t003:** Comparison of the parameters of similar microgrippers.

Microgripper	Single Side Amplification	Total Amplification
Reference [24]	–	22.89
Reference [25]	–	6.88
Reference [26]	–	16.4
Reference [27]	–	22.6
Reference [28]	3.68	3.68
Reference [29]	4.16	4.16
Reference [30]	6.1	6.1
Reference [31]	14.94	14.94
Design of this paper	31.6	31.6

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
