# Peer review of "Design of a Compliant Mechanism Based Four-Stage Amplification Piezoelectric-Driven Asymmetric Microgripper"

_micromachines, 2019, doi:10.3390/mi11010025_

Round 1

Reviewer 1 Report

Comments and Suggestions for Authors

Review Micromachines-648053

Research on compliant mechanism-based four-stage amplification

piezoelectric-driven asymmetric microgripper

Xiaodong Chen, Siya Hu, Zilong Deng , Jinhai Gao, Xingjun Gao

To provide a solution to overcome the traditional constraints and drawbacks of symmetrical and asymmetrical microgrippers the authors present a modeling, optimization and experimental study of a new four-stage amplification asymmetric microgripper. The introduction section does provide enough material to appreciate fully the effort of the authors.

Generally, the research presentation is well organized. The results obtained are appropriately compared with the similar designs in the performance comparison section.

Nevertheless, the authors should correct language errors and improve the narrative of the text to make it more comprehensible, concise and clear. There are also technical deficiencies that should be addressed. Improving suggestions are listed below:

Line

Comment

55

Consider the phrase correction“when the gripping object is a chip or other thin…”

58

Add to the phrase “In case of symmetrical gripping it is easy…” or “For the symmetrical gripping is easy…”

64

Insert the word “that” between “of lever amplification” and “can realize”

71

Mind the capital letter in reference surname Sun.

89

Check the decimal point in the sizes.

103

Check whether the first phrase is correct/necessary “Analyze the bridge type amplifier”. It is recommended to eliminate it or to avoid imperative tense.

103

Correct “amplificer”

104

Check whether the phrase is correct/necessary “Take one side length for stress analysis”. If so, try to avoid imperative tense.

105

The same situation as in 103 and 104

110

Check the quality of arrows, the black filler seems to be shifted.

111

Mind the capital letter in “Single”

114

See 105,104,103 – Avoid imperatives.

143

Consider the phrase revision: “the deformation of the flexure hinge is micro deformation”

159, 160

Mind the capital letters in the Figure and Table Titles.

171

Mind the gap “Fig . 6(b)”

172

Mind the gap “material ,”

176

Remove the indent at the beginning. The paragraph is directly related to the previous sentence, and there is no need to separate them in this way.

178

The last phrase is considered as redundant “When the microaxis…”, because Figure 7 presents the information explicitly. Instead, some details on the optimization procedure itself would be quite interesting. Even if you have already specified at the beginning of the 3.1. Size parameter Optimization, that the optimal geometric parameters obtained are based on the optimization of the structural dimension of the microgripper hinge, for readers would be quite interesting some technical details of the optimization procedure you’ve applied, in case if somebody would like to reproduce it.  Also, it would be quite reasonable to proceed following the order outlined in the abstract (see lines 21-22). Due to the fundamental importance of the structure optimization procedure, it would be better to provide it in the clearest way possible.

198

If possible specify the model of the isolation platform as well.

203

Eliminate the end of the phrase:” to verify the performance of the microgripper”

203-204

Eliminate the phrase: “A series of experiments…”

205

The term “theoretical working diagram” is better to be replaced by “functional block diagram”

222

Correct “has a stable performance”

225

Ti would be better to preserve the same terminology change “magnification” to “amplification”

233, 236

Remove the point after “13” and “14”

235

Please consider that the term self-adaptability is not sufficiently justified in this case. Even if we are talking about the automatic control of the force exerted, which is not the case here, we are still far away from the self-adaptability. What do we have is just a rigid contour of the gripping jaws capable to grasp differently shaped objects, but, to give you an example, not capable to change the shape of the grasping part according to the object in play. Thus, it is recommended to eliminate the word “self-adaptability” and reformulate the phrase accordingly.

239, 243

Unify the text styles and size.

244

Change “d” to “c”

251-253

Consider phrase revision

Author Response

    Thank you very much for your valuable comments and detailed description of the problems in the article, which is very helpful to improve the quality of the article. Thank you sincerely.

1. All questions shall be changed by the "review" function in "word".
2. The original text shall be modified as required and marked in red.
3. Delete some phrases as required.

Reviewer 2 Report

The submitted paper addresses the interesting and with regard to the journal “Micromachines” suitable topic of the design, simulation and test of an asymmetric piezo-driven flexure hinge-based compliant microgripper. Although the idea of using an asymmetric structure to ensure stable grasping could have potential and the paper content may be of relevance, the contribution of the paper and the impact of the presented compliant microgripper is not clear due to the fact, that the paper is not well readable (e.g. English writing, grammar and expressions, lots of typos and so on…), is not fully understandable/self-explanatory from content point of view (incompletions, missing or wrong symbols in the kinematic scheme) and comprises too less content regarding important details (influence of the lateral motion of the double parallel mechanism and the coupler rotation on the gripping precision, influence of the contour-dependent rotational axis shift on the overall motion behavior, selecting of the geometrical parameters, objective and detailed settings of the optimization procedure, detailed settings for the FEM simulation, measurement process, what is the really new contribution beside a high transmission ratio? I think the parallel gripping of the presented sample objects would be also possible with a symmetric microgripper…).

The overall paper is not really well prepared which concerns much formal aspects and the presentation of content as well as the quality and quantity of content. This begins with the fact, that a with the term “Research” in the title a very general formulation is chosen. It would be better to directly write what is the object or contribution of the paper. To the reviewer’s point of view, the motivation (state of the art for compliant mechanisms and of optimized flexure hinge shapes), the design approach, and the results/conclusion of the paper are not clear. The major part of the paper is about the design and the calculation of the transmission ratio of microgrippers, which has been thoroughly studied in literature. Therefore, the paper lacks of its own contribution. In addition, to many important terms are mixed together so that their use is uncommon, wrong or does not make sense (e.g. flexible mechanism instead of compliant mechanism, straight circular flexure hinge instead of asymmetric right-circular flexure hinge, straight flexible hinge=straight flexure hinge instead of corner-filleted flexure hinge, …). Then, the paper contains obvious content errors or contradictions that normally are not allowed to happen in such a high-ranked journal (e.g. strain is not stress!). Also lots of formal errors (e.g. text fragments, missing punctuation marks after equations in a sentence, missing spaces between values and units, …) are made. Then, the paper contains to many typos and the grammar as well as English spelling is poor. Due to this, the results are questionable because not comprehensive. Furthermore, the mentioned references are not sufficient (more the quality than the number, e.g. references for compliant mechanisms and flexure hinges are missing at all). In conclusion, the paper seems to be more a rush work then a well prepared comprehensive and concise paper in the context of this journal. Thus, considering the state of the art and current research output as well as the necessary high qualitative and quantitative standards for a well-prepared scientific-sound journal paper, the paper is not acceptable in the submitted form.

Additional concerns and comments:

1.) Line 2: What means research?

2.) Line 82: What means higher precision?

3.) Line 105: I think figure 3 does not show a stress analysis diagram, this term is used wrong.

4.) Figure 2: A fixed support of the parallel crank is missing. Why no hinges are model in the double amplifier mechanism? The lateral motion of the top hinge of the lever is not shown.

5.) Line 154: What hinge or hinges?

6.) Section 3.1: The optimization settings and objective are missing at all. This is not possible. What minimum notch height have all the hinges?

7.) Figure 5 contains only redundant information (cf. Figure 1) of the geometry of the microgripper and the numbering/letters could be included in Figure 1 too.

8.) The detailed FEM settings are missing at all. What are the settings concerning element type, mesh, the usage of mid-nodes, exact boundary conditions and loads (with a picture of selected geometry elements), load steps, convergence settings? Is a linear or nonlinear analysis performed (setting “nlgeom” on or of/non-linear geometry) and how is the influence of this setting in the context of the motion imprecision based on the inherent axis shift of the flexure hinges regarding a microgripper?

9.) Line 171: In figure 6b the stress is modeled and the strain is not calculated in MPa.

10.) Line 174: How is the grasping force analyzed and modeled in detail?

11.) Line 177: How is it determined that the grasping is stable?

12.) Section 4.2: How is the detailed measurement procedure including number of measurements, standard deviation and error discussion.

13.) The “Conclusion” section contains no really conclusions of results and it is more a summary of the working approach, which us not common.

Author Response

Thank you very much for your valuable comments and detailed description of the problems in the article, which is very helpful to improve the quality of the article. Thank you sincerely.

a. All questions shall be changed with "word" and "review" functions.
b. The original text shall be modified as required and marked in blue.
c. Some words are changed.
d. English has been polished and change punctuation

The main contributions of this paper are as follows:
A kind of four-stage amplification mechanism is designed and applied to the microgripper. According to the existing literature, only three-stage amplification microgripper is designed.The four-stage amplification microgripper designed in this paper is the first four-stage amplification mechanism. .
why the former did not design the four-stage amplification mechanism:

Among the many documents consulted, the single side maximum amplification of the two-stage amplification mechanism is not more than 10 times, and the single side maximum amplification of the three-stage amplification mechanism is not more than 12 times, further research  to improve the magnification is needed. However, the maximum amplification of the single side of the four-stage amplification mechanism designed in this paper is 31.6 times. This mainly depends on the series connection of two bridge amplifier , which effectively improves the amplification.

There are the responds to the questions:

1. Replace "research" with "design";
2. After careful consideration, the word "higher precision" has been removed.
3. The statement has been changed.
4. Figure 2 has been changed as required.
5. The singular and plural questions have been changed
6. Your opinion is very important to improve the quality of the article. We have written the detailed steps of optimization as required. The minimum gap is 0.64mm.
7. We have a serious consideration on this issue. Your opinion is very good. It is very reasonable to combine the two figures together.
8. Grid, load, boundary condition, linear analysis and other problems have been added in the original text as required, and "Fig. 5 (a)" has been added.
9. "Fig. 6 (b)" is the stress diagram and has been changed.
10. We have a very serious consideration on this issue. Your comments are very good and have been changed as required.
11. Remove "stable" and replace it with the linearity of the input display and the output force is large and easy to control.
12. We have a serious consideration on this issue, and your opinion is very good. Add Table 2 for the number of measurements, and discuss the standard deviation and error in detail.
13. Some changes have been made to the conclusion.

Round 2

Reviewer 2 Report

The paper has been revised, but unfortunately most of the most important points in the text of my review (two first paragraphs) have not been addressed at all. The 13 points at the end of my review were only some additional (minor) comments. Thus, also the revision seems to be a quick work and a major revision is again necessary before the paper can been accepted.

Comments:

1.) Please check again English writing, avoid typos and especially proof settings of variables, indices, spacings after numbers and before units, also of the new added parts of the paper.

2.) Please answer the question, if parallel gripping of the presented sample objects would be also possible with a symmetric microgripper and when not why?

3.) Please include your additional comments (in the author’s response) regarding the new contribution of the paper also in the introduction to some extent.

4.) What is the influence of the lateral parasitic motion of the coupler of the double parallel mechanism and the coupler rotation (for the compliant mechanism only) on the gripping precision? Perhaps include this issue in the drawing of Fig. 4 and the discussion too. Please read in detail and refer to some of the following references:

a) Awtar, S.; Slocum, A. H. (2005): Closed-form nonlinear analysis of beam-based flexure modules. in: Proceedings of IDETC/CIE 2005. ASME 2005 International Design Engineering Technical Conferences & Computers and Information in Engineering Conference. Long Beach, California, USA. DOI: 10.1115/DETC2005-85440.

b) Luo, Y.; Liu, W.; Wu, L. (2015): Analysis of the displacement of lumped compliant parallel-guiding mechanism considering parasitic rotation and deflection on the guiding plate and rigid beams. in: Mechanism and Machine Theory 91, p. 50–68. DOI: 10.1016/j.mechmachtheory.2015.04.007.

c) Gräser, P.; Linß, S.; Zentner, L.; Theska, R. (2017): Design and Experimental Characterization of a Flexure Hinge-Based Parallel Four-Bar Mechanism for Precision Guides. in: Microactuators and Micromechanisms, 45. Cham: Springer International Publishing (Mechanisms and Machine Science 45), p. 139–152. DOI 10.1007/978-3-319-45387-3_13.

5.) What is the influence of the notch contour-dependent rotational axis shift of each flexure hinge on the overall motion behavior? Please read in detail and refer to some of the following references:

d) Zelenika, S.; Munteanu, M. G.; Bona, F. De (2009): Optimized flexural hinge shapes for microsystems and high-precision applications. In: Mechanism and Machine Theory 44 (10), pp. 1826–1839. DOI: 10.1016/j.mechmachtheory.2009.03.007.

e) Linß, S.; Schorr, P.; Zentner, L. (2017): General design equations for the rotational stiffness, maximal angular deflection and rotational precision of various notch flexure hinges. In: Sci. 8 (1), pp. 29–49. DOI: 10.5194/ms-8-29-2017.

f) Valentini, P. P. and Pennestrì, E. (2017): Elasto-kinematic comparison of flexure hinges undergoing large displacement, Mechanism and Machine Theory, 110, 50–60, DOI: 10.1016/j.mechmachtheory.2016.12.006.

6.) Fig. 2 is not sufficient at all because nothing can be seen from it. Please revise it totally with appropriate representations and line widths for example so that it is self-explanatory. Additionally, lots of arrows are at the false position. Please revise the caption

7.) Table 1: What dimension represents the flexure hinge width in your case, the minimum notch height or the dimension in z-direction? Thus, also an additional drawing where the dimensions and parameters of the flexure hinges are visible and a coordinate system in Fig. 1 is missing perhaps.

8.) In my point of view one very critical point regarding the investigation of a high-precision motion system, which is based on compliant mechanisms incorporation flexure hinges, is that “only” a geometrically linear FEM analysis is performed. My experiences are that the difference of the motion behavior compared to a nonlinear analysis are very large (in the micrometer range) and thus perhaps relevant in your case. Perhaps this point could be checked?

9.) Again the question, how is the grasping force analyzed and modeled in detail via the FEM simulation? I have found no changes/comments regarding this.

10.) Please revise and adjust the caption of Fig. 5.

Author Response

Dear Editor and reviewers,

Thank you very much for examining my paper and for providing these valuable comments. The paper has been revised thoroughly based on the comments and suggestions, with main changes highlighted in red. Please find the following detailed responses to each point raised by reviewers.

Thank you very much for your consideration.

Yours sincerely,

Xiaodong Chen

Comments and Responses

Please check again English writing, avoid typos and especially proof settings of variables, indices, spacings after numbers and before units, also of the new added parts of the paper.

The paper has been revised thoroughly based on these comments and suggestions.

Please answer the question, if parallel gripping of the presented sample objects would be also possible with a symmetric microgripper and when not why?

It is also possible to realize the parallel grapping for holding the presented sample objects using a symmetric microgripper. However, the proposed asymmetric microgripper has more stable grasping forces. The proposed compliant mechanism can also be employed to design a symmetric microgripper with the same displacement amplification, while control of the grasping forces is more complicated and the geometric dimension is much larger.

The above responses are added in the revised paper.

Please include your additional comments (in the author’s response) regarding the new contribution of the paper also in the introduction to some extent.

The comments have been included in the revised paper.

What is the influence of the lateral parasitic motion of the coupler of the double parallel mechanism and the coupler rotation (for the compliant mechanism only) on the gripping precision? Perhaps include this issue in the drawing of Fig. 4 and the discussion too. Please read in detail and refer to some of the following references:

The output stage of the compliant parallelogram parallel mechanism on the moveable jaw has a parasitic rotation about the Z-axis, which reduces the grapping accuracy of the jaw. However, the parasitic rotation is much smaller than the principal motion, the translation along the X-axis. It can be calculated from finite element analysis that the rotation displacement is about 0.0658% of the gripping displacement, as shown in Figure 4. Therefore, the parasitic rotation can be ignored in this case, while it can also be eliminated by applying the actuation force along the stiffness center of the compliant parallelogram parallel mechanism (shown in Fig. 4(b)) in other cases if larger geometric dimension is not a problem.

The above responses are added in the revised paper, and the suggested papers are also cited in the revised paper.

(a)                             (b)

Figure 4. Amplification principle of parallelogram mechanism

What is the influence of the notch contour-dependent rotational axis shift of each flexure hinge on the overall motion behavior? Please read in detail and refer to some of the following references:

The short sheet beam rotational joints have larger rotational axis shift compared with the notch joints, but the short sheet beam rotational joints can offer larger motion ranges with smaller stress concentration. The relevant comments have already been added in the revised paper. In addition, the suggested papers are also cited in the revised paper.

2 is not sufficient at all because nothing can be seen from it. Please revise it totally with appropriate representations and line widths for example so that it is self-explanatory. Additionally, lots of arrows are at the false position. Please revise the caption

Figure 2 have been revised based on the comments.

Table 1: What dimension represents the flexure hinge width in your case, the minimum notch height or the dimension in z-direction? Thus, also an additional drawing where the dimensions and parameters of the flexure hinges are visible and a coordinate system in Fig. 1 is missing perhaps.

The paper has been revised based the comments.

In my point of view one very critical point regarding the investigation of a high-precision motion system, which is based on compliant mechanisms incorporation flexure hinges, is that “only” a geometrically linear FEM analysis is performed. My experiences are that the difference of the motion behavior compared to a nonlinear analysis are very large (in the micrometer range) and thus perhaps relevant in your case. Perhaps this point could be checked?

The simulation analysis uses the static analysis of the workbench module of ANSYS to analyze the model in detail. The deviation between static linearity and static nonlinearity is very small because of the small motion ranges.

Again the question, how is the grasping force analyzed and modeled in detail via the FEM simulation? I have found no changes/comments regarding this.

There are two steps (first constrain the fixing holes): The first step is to get the output force of piezoelectric ceramics through the deformation relationship. The nominal displacement is applied at the input end of the microgripper, the corresponding output end produces a certain amount of deformation, and the deformation is recorded. Then measure again, apply a certain force on the input end of the microgripper, the corresponding output end has the corresponding output displacement, and get the relationship between the input force and the output displacement through the proportional relationship. The second step is to get the output force of piezoelectric ceramics and the gripping force of microgripper through the strain relation, In grid state. First, the output force (as above) is obtained by strain relation, and a certain force is applied at the input end to specify a point of the grasping jaw to obtain the strain size of the point; then the position of the applied force is changed to the position where the grasping jaw is connected with the hinge, continue to check the strain size, and so on, to obtain multiple FEA gripping force points.

Please revise and adjust the caption of Fig. 5.

The title of Figure 5 has been revised.

Round 3

Reviewer 2 Report

The authors have revised the paper or answered the questions according to the reviewer’s comments no. 1-8 and no. 10. These provided responses are satisfactory and the paper can be accepted for publication after a minor revision according to some necessary changes: Please check again correct English writing of the new added text, there are still some grammar problems. Then, please answer question 9 shortly in the paper too. How is the grasping force of the gripping jaws analyzed and calculated in detail with ANSYS? This is a tricky point when you want to do it right and not only as first relatively simple approximation. Please also give information about the assumptions made here. Is the simulation done with or without gripped object. Because to the reviewer’s knowledge the gripping output force of a monolithic compliant two-finger gripper strongly depends on the object size (X), width (Y), and stiffness (ideally rigid or deformable) as well as the size of the gripping surface. Then perhaps the simulation with an object based on the contact forces during the gripping process would be necessary regarding accurate results.

Author Response

Dear Editor and reviewers,

Thank you very much for examining my paper and for providing these valuable comments. The paper has been revised thoroughly based on the comments and suggestions, with main changes highlighted in red. Please find the following detailed responses to each point raised by reviewers.

Thank you very much for your consideration.

Yours sincerely,

Xiaodong Chen

The authors have revised the paper or answered the questions according to the reviewer’s comments no. 1-8 and no. 10. These provided responses are satisfactory and the paper can be accepted for publication after a minor revision according to some necessary changes: Please check again correct English writing of the new added text, there are still some grammar problems.

The English writing of this paper has been checked and polished.

Then, please answer question 9 shortly in the paper too. How is the grasping force of the gripping jaws analyzed and calculated in detail with ANSYS? This is a tricky point when you want to do it right and not only as first relatively simple approximation. Please also give information about the assumptions made here. Is the simulation done with or without gripped object. Because to the reviewer’s knowledge the gripping output force of a monolithic compliant two-finger gripper strongly depends on the object size (X), width (Y), and stiffness (ideally rigid or deformable) as well as the size of the gripping surface. Then perhaps the simulation with an object based on the contact forces during the gripping process would be necessary regarding accurate results.

This paper proposes a new compliant mechanism based gripper, with advantages of large motion amplification ratio and operation stable performance compared with the existing designs. The modelling and FEA simulations in this paper aims to analysis the motion amplification ratio and the force-displacement relationship between the input and the output.

The exact gripping force of the monolithic compliant gripper depends on the stiffness of the gripper, the object dimension and stiffness, as well as the size of the gripping surface. However, this paper mainly focuses on the stiffness of the gripper between the input and the output, which can reflect the maximum gripping forces that the gripper can offer. The gripping forces mentioned in this paper actually are the gripping forces that the gripper can provide. In order to make it more clear, the “gripping force of the gripper” is changed by “output force of the gripper” in the revised paper.

The effects of the object dimension, stiffness and the size of the gripping surface on the gripping force are not addressed in this paper, but it will be a very good future work for our studies.

This manuscript is a resubmission of an earlier submission. The following is a list of the peer review reports and author responses from that submission.